



# Differences and influencing factors for underground water carbon uptake by karsts in Houzhai Basin, southwest China

Junyi Zhang[1,2], Zihao Bian[1], Minghong Dai[1] , Lachun Wang[1]*, Weici Su[3]

1.School of Geographic and Oceanographic Sciences, Nanjing University, Nanjing 210023, China.

2.School of Tourism and Land Resources, Chongqing Technology and Business University, Chongqing 400067, China.

3.The Institute of Mountain Resources, Guizhou Academy of Sciences, Guiyang 550018, China.

Correspondence to: Lachun Wang (wang6312@263.net.cn)

**Abstract**

Karst geological carbon sink is an important part of the global carbon sink, so how to get the accurate carbon sink of karst ecosystem has become the core issue of the research. We used flow and carbon ion concentration data from three stations with different environmental background conditions in the Houzhai basin to analyze the differences in carbon uptake between stations and their impact factors. Results show that carbon sink discharge was mainly controlled by the flow of each site. The rapid increase in flow only has a partial dilution effect on ion concentrations, preliminary analysis considered due to the high speed and stability of chemical carbonate weathering. LUCC type has important effects on the bicarbonate ions concentrations, if runoff is stable, the influence of flow variation on ion concentration will be less than the effects of chemical carbonate weathering in different environmental conditions (comparison of Laoheitan and Liugu station results is 150%) on bicarbonate ion concentrations. However, if runoff increases significantly, the impact of runoff variation on bicarbonate ions will be greater than the effects of chemical carbonate weathering by different environmental conditions (comparison results of Laoheitan and Maoshuikeng station). This work provides a reference for the calculation of karst geological carbon sink.

## 1. Introduction

Global warming from emissions of greenhouse gases has become the core issue of global environmental change. One of the most pressing concerns in the science of global climate change is the effective accounting of the global budget for atmospheric $CO_2$ (Schindler,1999;Melnikov and Neill,2006;Liu et al.,2010;Kao et al.,2014), since in order to control global warming, it is necessary to control emissions of carbon dioxide through carbon capture and storage (CCS) technology. In addition to developing CCS technology, an understanding of a number of natural ecological and geological processes such as rock weathering, plant growth, and other physical, chemical, and biological processes can also improve CCS (Hoffmann et al.,2013). Carbonate weathering in rock weathering processes is considered to be both an important source and sink of $CO_2$ (Zeng et al.,2015;Lian et al.,2011;Liu and Zhao,2000;Serrano-Ortiz et al.,2010;James et



al.,2006). Carbonate rock dissolves more easily in water in which $CO_2$ is dissolved, and at a
temperature of 15 ℃ and atmospheric $CO_2$ partial pressure of 380 ppmv, the equilibrium
concentration of the dissolved inorganic carbon (DIC) in a water system of $CaCO_3$–$CO_2$–$H_2O$ can
reach 1231 mol/L in the water with calcium carbonate (Dreybolt,1988). Moreover, karst is widely
distributed around the world; it occupies about 11.2% of the Earth's surface, and about 15 million
$km^2$ in the earth (Dürr et al.,2005). Therefore, carbonate is closely associated with atmospheric
$CO_2$ concentrations through carbonate weathering processes and becomes an important component
of the global carbon cycle. As a result, carbon uptake from chemical weathering can significantly
influence the evolution of atmospheric [$CO_2$] in the Earth's long-term (over the past 100 million
years) (Berner et al.,1983) and short-term climate (Liu,2012). Moreover, the previous research has
shown that more carbon is sequestered from carbonate weathering than from silicate rock
weathering (Liu,2012).

48       Consequently, it is very important to accurately estimate net carbon uptake from carbonate

weathering processes. Currently, there are two main methods for calculating carbonate weathering
carbon sinks. The first method uses the empirical relationship between carbon uptake rates and
different lithology types and calculates the weathering by determining the different empirical
dissolved constants such as 0.0294 (g C $mm^{-1}$) or 0.0383g C $mm^{-1}$ estimated by Amiotte Suchet
and Probst (1995) and Bluth and Kump (1994), respectively. The other method estimates carbon
sinks using observations of river chemistry such as karst water flow and concentrations of
bicarbonate. Nevertheless, there are always some differences between the results of the two
calculation methods (Yan et al.,2011).

57       Karst is widely distributed in China, which has approximately 3.44 million $km^2$ of karst area,

including buried, covered, and exposed carbonate rock areas (Jiang et al.,2014), and about 0.4
million $km^2$ of karst is located in the southwest (Jiang and Yuan,1999). The most frequently used
calculation methods for carbon sequestration are the forward method (Zhang,2011) and, in
China's karst regions, the river chemistry method. But, there are some defects of the forward
method because physical models cannot truly reflect the in situ karstification and carbon migration
process (Zhiqiang et al.,2011), and for this reason the river chemistry method is more frequently
adopted (Zhao et al.,2010;Yan et al.,2012;Zhang et al.,2015;Huang et al.,2015).

65       There are large discrepancies in the estimates of carbon sequestration in China, ranging from

5 Tg $Cyr^{-1}$ (Jiang and Yuan,1999) to 12 Tg $Cyr^{-1}$ (Yan et al.,2011) and18 Tg $Cyr^{-1}$ (Liu and
Zhao,2000). These values are usually derived from the observed carbon discharge from a single
water chemical observatory in a single basin in southwest China; however, there may be some
deviations in the results of this single observation site because of the high heterogeneity of the
karst system, the sensitivity of the response to external environment changes, and the interference
of human activity which is usually intensified in karst regions. Studies have shown that, carbonate
weathering is sensitive to ecosystem dynamics, which means that carbonate weathering and
associated $CO_2$ consumption discharges quickly react to any global changes or land use



modifications (Calmels et al.,2014). Therefore, in this study we used flow and carbon ion
concentration data from three observation stations with different environmental background
conditions in the same karst groundwater basin in order to analyze the differences in carbon
uptake between stations and their impact factors. This work also provides a reference for
improving the calculation accuracy of karst geological carbon sink.
**2. Materials and methods**
*2.1. Study Area*
Houzhai basin is located in Puding county in the middle of Guizhou province (26 °13 '- 26 °
15' N, 105 °41 '105 °43' E). The total area of the basin is 80.65 km2, and the length of the main
river is about 12 km (including the ground and underground river) (Figure 1). The southeastern
portion of the basin is lower than the northwestern portion. The relative elevation of the basin is
about 150 m, and its average altitude is 1250 m. A typical hoodoo depression physiognomy is
distributed in the east of the basin where the main land-use type is forest vegetation, while karst is
distributed in the west of the basin where the main land-use type is farmland. It has a subtropical
humid climate; the average rainfall is 1316.8 mm and the average temperature is 15.5 ℃. The
rainy season occurs from May to October and the dry season from November to April.
Precipitation during the rainy season accounts for more than 80% of annual rainfall. Bedrock in
the basin is composed of mainly carbonate rock formed during the Triassic. As a result of
lithology and geological structure, karstification is strong and karst formation is widely developed
in the basin. Hydrological runoff processes are significantly influenced by karst underground
space (gap and pipe) and its distribution characteristics. There is no obvious surface river valley
upstream, and although there is a river valley midstream and downstream, seasonal runoff only
appears temporarily, and leakage pits are arranged along the riverbed.

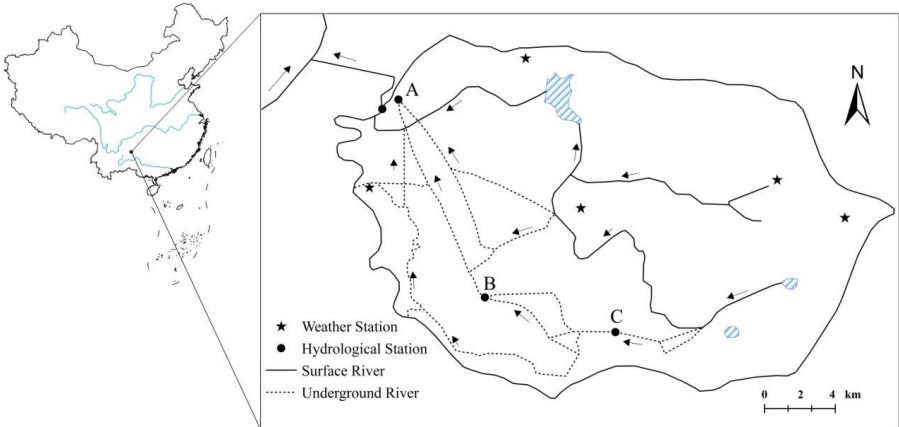


Figure 1. The distribution of drainage systems and weather hydrological stations.
*2.2. Date Sources and Methods*
*2.2.1. Date Sources*
The main data are derived from three groundwater hydrology and water quality monitoring





stations: A (Maoshuikeng MSK), B (Liugu LG), and C (Laoheitan LHT) (Figure1). The three

stations are located in the upstream, midstream and downstream reaches of the basin, respectively.

The LHT station is located on the edge of the peak cluster depression, the LG station is located in

the region of the peak cluster basin, and the control areas of the two stations are 24.06 km$^2$ and

15.81 km$^2$, respectively (Wang et al.,2010). MSK station is located at the outlet of the

underground river, and its control area is 80.56 km$^2$ (Figure1). We selected continuous and

complete data, which contain the average daily flow data and $HCO_3^-$ concentration data from

MSK station (1996-2001), LG station (1992-1996), and LHT station (1988-2002). Average annual

temperatures from 1988-2002 were provided by Puding station, which was located at the boundary

of the basin. Concentration data were measured directly by the water samples station. Water

samples were collected from the underground rivers at a water depth of 0.6 m at exit, 6 times per

month in the rainy season (May to October) and 3 times a month in the dry season (November to

April of the following year) sampling, and water samples were measured for pH using a portable

meter. Water temperature and the concentration of bicarbonate ([$HCO_3^-$]) were determined by

titration with standard hydrochloric acid (HCl) immediately after samples were taken at the

sampling site.

### 2.2.2. Determination of Water Samples and DIC Method

Bicarbonate concentration was measured using a neutralization titration method. The steps of

the method are as follows: (1) Add a sample to a 100 ml beaker and drip 4 drops of phenol red

indicator and then shake the sample well; (2) Titrate the sample using standard HCl (0.025 mol/L)

until the red disappears at a pH of 8.4 and record the standard HCL usage quantity $V1$; (3) Drip

three drops of methyl orange indicator into the sample and shake well, then titrate using standard

HCl until the color of the sample changes to orange at a pH of 4.4, and record the HCl usage

quantity ($V2$); and (4) Finally, measure the concentration of carbonate ions in the water samples

by using formula (1):

$$\rho = \frac{(V_2 - V_1) \times c \times 61.017 \times 1000}{V} \qquad (1)$$

In a karst environment, carbon dioxide dissolves in water and undergoes a reversible chemical

process (2) with calcium carbonate:

$$CaCO_3 + H_2O + CO_2 \Leftrightarrow Ca^{2+} + 2HCO_3^- \qquad (2)$$

Under a steady state, the quantity of carbon dioxide dissolved in karst water is equal to the

discharge of CO$_2$ from the atmosphere. That discharge in g C m$^{-2}$ time step$^{-1}$ is calculated

according to the following formula (3) (Yan et al.,2011;Amiotte-Suchet and Probst,1993).

$$F = \frac{1}{2} cq \frac{M_c}{M_{HCO_3}} \qquad (3)$$



where c is the concentration of bicarbonate ions($g/m^3$); q is the production flow ($m^3$/time
step); $M_C$ and $M_{HCO3}$ are the molecular weights of C and $HCO_3^-$, respectively, and 1/2 means that
1 mol of bicarbonate needs only half a mole of $CO_2$ from the soil or atmosphere. Additionally,
karst water is generally alkaline. The content of $CO_3^{2-}$ C in dissolved inorganic C is very small, so
we did not need to consider it in the DIC calculation (Gelbrecht et al.,1998;Yan et al.,2011). In
this study, we used the formula $F_1$ below to calculate net carbon uptake by karst, using the
estimates of year mean [HCO3$^-$], ion concentration during the dry-wet season, and the mean daily
underground flow discharge.

$$F_1 = \frac{1}{2} \cdot \frac{M_C}{M_{HCO_3}} \cdot \bar{c} \cdot \sum_{n=1}^{12} q_n \qquad\qquad (4)$$

where $\bar{c}$ is either the annual average bicarbonate density or the ion concentration in the dry-wet
season (mg/L), and q is the average daily excretion( $m^3$/s, n=365 day).

## 3. Results

### 3.1. Dry -Wet Seasonal and Inter-annual Variations of Ion Concentration and Discharge

For each site during the study period, the ion concentration in the wet season was slightly
smaller than in the dry season. LHT station, which had the longest study period, exhibited the
highest and lowest values for bicarbonate ion concentration, which were 240.5mg/L (1994) and
201.7 mg/L (1999) in the rainy season, 259.6 mg/L (2002) and 234.7mg/L (1991) in the dry
season, and 248.3 mg/L (1994) and 218.8 mg/L (1999) for the whole year, respectively. Moreover,
there was a negative correlation between ion concentration and discharge (Figures 2, 3, and 4).
From 1992-1996, the annual average concentration of bicarbonate ions in the rainy season, dry
season, and whole year were 228.8 mg/L, 249.3 mg/L, 239.1 mg/L, respectively for LHT station
and 222.0 mg/L, 253.5 mg/L, 237.8 mg/L, respectively at LG station. Although there is little
difference in ion concentration between the two stations, when considering the stability of ion
concentration changes (Table 1), LG station was more stable than LHT station. During the same
period, from 1996-2001, the annual average ion concentrations in the rainy season, dry season,
and whole year were 217.8 mg/L, 247.4 mg/L, and 232.6 mg/L, respectively for LHT station, and
209.9 mg/L, 226.4 mg/L, 218.2 mg/L, respectively for MSK station. Table 1 shows that MSK
station was more stable than LHT station with respect to the standard deviation of ion
concentration variation. Although, the difference in ion concentrations between LHT and LG was
smaller than that between LHT and MSK, differences in the site as a whole were small.
The discharge from MSK station, which is located at the outlet of the underground river basin,
was larger than the discharge from LG and LHT. From 1996-2001, the annual average flow values
of MSK in the rainy season, dry season, and whole year were 282.5 $m^3$, 121.3 $m^3$, 403.9$m^3$,
respectively, and the flow in the rainy season was significantly greater than in the dry season. The
flow of LG and LHT in the rainy and dry seasons exhibited the same trend (Figure 2, 3, and 4).


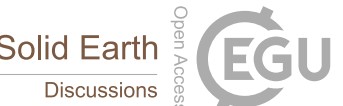

From Table 1, we can determine the stability of flow as follows: MSK>LG>LHT.
Table1. Standard deviation of production flow, ion concentration, and carbon sink for each station in the dry
and wet seasons and over the whole year

| Station | Water Discharge Rate | | | Bicarbonate Concentration | | | Carbon Uptake | | |
|---|---|---|---|---|---|---|---|---|---|
| | wet | dry | year | wet | dry | year | wet | dry | year |
| MSK | 19.59 | 19.87 | 21.83 | 6.46 | 6.64 | 5.26 | 1.12 | 0.84 | 0.43 |
| LG | 19.78 | 2.67 | 18.55 | 10.64 | 4.81 | 4.99 | 2.93 | 0.45 | 1.54 |
| LHT | 13.28 | 7.33 | 16.37 | 10.04 | 8.41 | 7.55 | 3.34 | 1.84 | 2.05 |


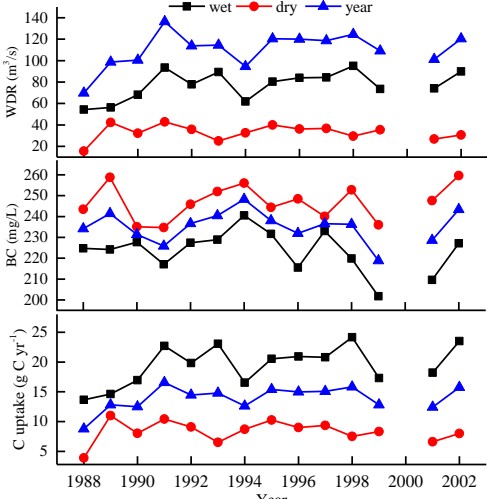

Figure 2. Variation in runoff, ion, and carbon sink for LHT station (1988-2002).

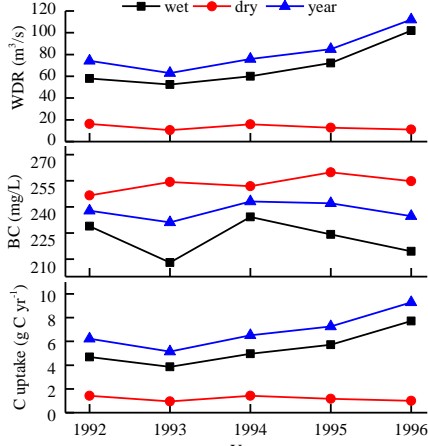

Figure 3. Variation in runoff, ion, and carbon sink for LG station (1992-1996).





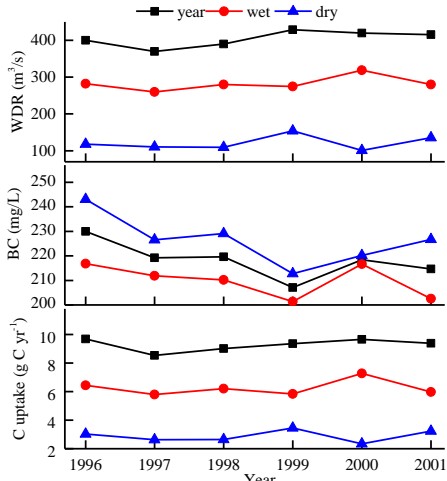


Figure 4. Variation in runoff, ion, and carbon sink for MSK station (1996-2001).

**3.2. Dry and Wet Seasonal and Inter-annual Variations in Carbon Uptake Rate**
From 1996-2001 at MSK station, the annual average carbon sink discharges of underground
water in the rainy season, dry season, and whole year was 12.51, 5.78, and $9.28 g/m^2$, respectively,
with a significantly greater net discharge in the rainy season compared to the dry season. From
1992-1996 at LHT station, the annual average carbon sink discharges in the rainy season, dry
season, and whole year was $10.78 g/m^2$, $2.38 g/m^2$, and $6.89 g/m^2$, respectively. From 1988-2002
(data from 2000 is missing) at LHT station, the annual average carbon sink discharges in the rainy
season, dry season, and whole year were 19.48, 8.34, and $13.91 g/m^2$, respectively, greater than
both LG and MSK stations (Figure 6). From 1996-2002 at LHT station, the annual average net
carbon sink discharges in the rainy season, dry season, and whole year were $20.82 \ g/m^2$, $8.14 \ g/m^2$,
and $14.48 \ g/m^2$, respectively while from 1992-2002 the respective values were $20.18 \ g/m^2$, 8.72
$g/m^2$, and $14.45 \ g/m^2$. Comparing the results for the same period, we found that the annual carbon
sink discharge in the rainy season, dry season, and whole year for LHT station were greater than
those for MSK and LG stations. However, with respect to the stability of carbon discharge (Table
1), MSK was the most stable in the rainy season while LG was the most stable in the dry season.
**4. Discussion**
**4.1. Flow and Ion Concentration Change and its Effects on Carbon Sink**
According to the flow trend of each station, we can see that the flow in the rainy season is
consistent with the flow trend for the whole year, suggesting that the runoff from precipitation in
the rainy season accounts for the majority of the annual runoff. This is mainly a result of the
monsoon climate where summer (May-September) precipitation levels are significantly higher
than in the winter (December-February); however, the flow trend in the dry season was smooth
(figure 5) because of less rainfall in the dry season when the runoff was mainly supplied by soil



water and fissure/pore water. It also suggests that the composition of underground karst aquifer
medium structures have important effects on the dry season flow. However, due to the difference
between the control area and the surface to underground diversion ratio, the flows between sites
cannot be compared.
The trends in annual runoff among sites are consistent with carbon sink discharge but differ
from the trends for bicarbonate ions (figure 2, 3, 4). This suggests that the effect of flow change on
carbon sink is greater than on ion concentration. According to changes in ion concentrations in the
rainy season, dry season, and whole year (figure 2, 3, and 4), flow correlated negatively with
carbon ion concentration, but if there was a significant difference between flow in the rainy season
and in the dry season, bicarbonate ion concentrations would not decrease when the flow increased
rapidly. Although we found differences between bicarbonate ion concentrations in the dry and wet
seasons (ion concentrations in the dry season are greater than in the rainy season), they were
small.
We then contrasted the results of each site. From 1992-1996, the annual average carbon ion
concentration was 237.8 mg/L for LG station and 239.1 mg/L for LHT station. The annual average
flow of LHT station was 1.37 times that of LG station, but the ion concentration did not decline
significantly due to the increase in flow. The basin area controlled by LHT station is characterized
by peaks and valleys, which have good vegetation cover that recovers rapidly. Previous studies
have shown that the concentration of $HCO_3^-$ is vulnerable to LUCC (land cover and land use
change) and other environmental changes (Zhao et al.,2010;Lan et al.,2015). In particular, the fast
recovery of vegetation can significantly promote the dissolution of carbonate and thus increase
bicarbonate ion concentration in karst groundwater (Liu et al.,2010;Berner,1997). This suggests
that when there is little change in flow, the effect of flow increase on ion concentration dilution is
smaller than the environmental effects of chemical carbonate weathering.
From 1996-2001, annual average carbon ion concentrations at LHT and MSK stations were
232.6 mg/L and 218.2 mg/L, respectively, but the average annual flow for LHT was only 115.6 $m^3$,
while MSK exhibited an annual flow of 3.49 times that value. Similarly, ion concentration did not
decline significantly as a result of the increase in flow. MSK station is located at the edge of a
paddy field, which has thicker soil coverage, and the underground rivers have more biological
carbon sources that could produce more $HCO_3^-$ in the ground water compared to LHT. Therefore,
the flow only has a partial dilution effect on ion concentration. Meanwhile, the effect of flow
increase on ion concentration dilution exceeded the environmental effects of carbonate weathering.
This shows that the dilution effects of the flow change on ion concentrations were not
multiplicative. That is to say, the flow was just a part of the dilution effect on ion concentrations,
and thus carbonate weathering was significantly affected by factors other than flow. Bicarbonate
ion concentrations of karst underground water may have a relatively stable extremum when

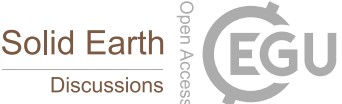

environmental conditions are stable. In addition, although studies have shown that under the
conditions of a stable LUCC, the strength of the carbon sink from rock weathering will depend on
the climate (e.g. temperature T, precipitation P) (Hagedorn and Cartwright,2009;Gislason et
al.,2009;Tipper et al.,2006), the annual average carbon sink trend for LHT station, which had the
longest study period (1988-2002), differed significantly from the annual average temperature trend.
However, this may be a result of time resolution limitations of the monitoring data.

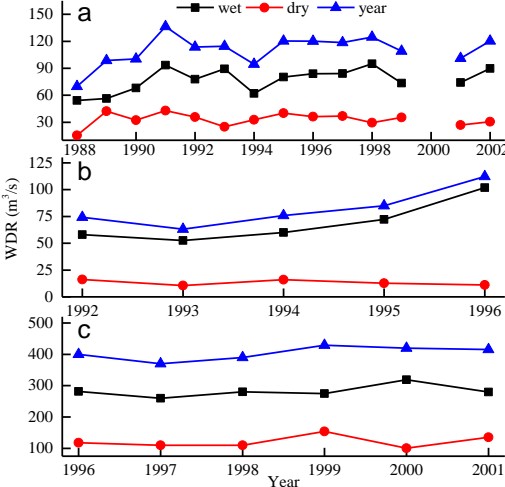


Figure 5. Variation in flow among sites in the rainy season, dry season, and whole year during the study period (a.
LHT; b. LG; c. MSK).

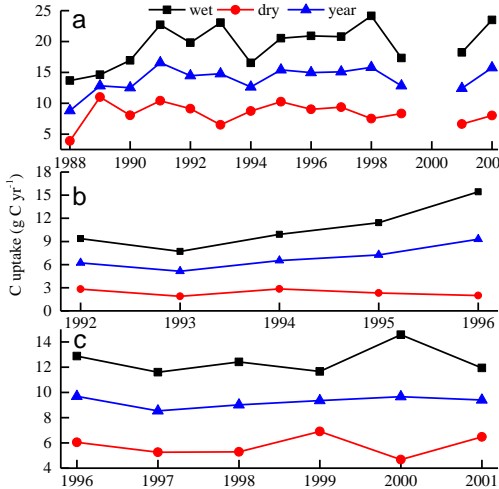


Figure 6. Variation in net carbon discharge among sites in the rainy season, dry season, and whole year during the
study period.





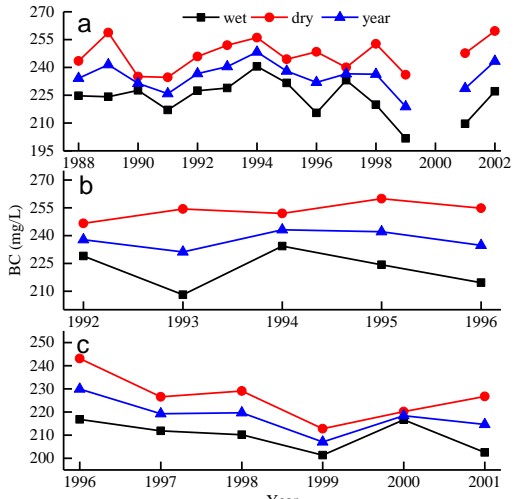


Figure 7. Variation in bicarbonate ion concentrations among sites in in the rainy season, dry season, and whole
year during the study period.
**4.2. Variation in Carbon Sink Discharge for Each Site**
The carbon sink discharge for each site in the rainy season was greater than for that of the dry
season and the annual average, while the carbon sink discharge in the dry season was less than the
annual average (figure 6). This shows that the karst carbon sink (karstification's absorption of
atmospheric $CO_2$ and soil $CO_2$) changes significantly with the seasons and exhibits striking
seasonal patterns. The reason for this is that the considerable summer rainfall runoff significantly
increases the amount of carbon sink discharge in the rainy season. The annual average carbon sink
discharge of all the stations during study period shows that LHT>MSK>LG; however,
comparisons cannot be made due to the different study periods.
During the same period, the carbon sink values for LHT station in the rainy season, dry
season, and whole year were greater than both LG and MSK stations. From 1992-1996, the flow
for LG in the rainy season and dry season were both significantly less than for LHT (figure 6). The
annual average concentration for LG in the rainy season and dry season were 239.1 mg/L and
237.8 mg/L, respectively, and are slightly less than LHT (figure 6). The difference in carbon sink
discharge between the two stations results from differences in flow. Furthermore, the LHT control
basin station is surrounded mainly by forest vegetation while the LG control basin is surrounded
mainly by dry farmland, and the different LUCC types may further increase differences in carbon
sink discharge between the two stations.
From 1996-2001, the carbon sink discharge for LHT station in the dry and wet seasons and
whole year were greater than for MSK station. The annual average concentration for LHT (230.4
mg/L) was greater than for MSK (218.2 mg/L), while the runoff was significantly greater for



MSK than for LHT. On the one hand, the fact that the carbon sink discharge for MSK was less
than LHT might be linked to the water conveying distance and LUCC type of the control area. The
carbon sink for MSK, which is the groundwater outlet for the whole basin, was influenced by the
landform and LUCC type of the entire river basin (Figure 1). Previous research has shown that
karst erosion rates under soil vary significantly for different LUCC types in karst watersheds, and
the averages for cultivated land, thickets, secondary forests, grassland, and forest were found to be
4.02, 7.0, 40.0, 20.0 and 63.5 t $km^2$ $a^{-1}$, respectively (Zhang,2011), with the erosion rate of
carbonate karst under the cover of cultivated land being the lowest (Yan et al.,2014). Previous
research has also shown that vegetation can increase the speed of weathering by 3-10 times
(Berner,1997). According to monitoring data from Guilin province in China vegetation restoration
can significantly increase the average annual concentration of soil $CO_2$ (increased by 266% in 10
years). The increase in $CO_2$ promotes the dissolution of carbonate rock and greatly increases
$HCO_3$ concentrations in groundwater (Liu,2012;Waterson and Canuel,2008). Research in the
Houzhai valley has shown that forest recovery causes more carbon dioxide ($CO_2$) to be dissolved
in karst water, which in turn allows for carbon uptake by forests (Yan et al.,2014). This research
also suggested that karst hydro-geochemistry and the karst-related carbon cycle could be regulated
effectively by different LUCC types (Zhao et al.,2010). On the other hand, in the process of runoff
converging at the outlet, much of the water flows into the surface river and flows across the thick
soil of paddy fields, but our calculation method only considers carbonate weathering carbon sinks
(water - rock - gas interaction) and not the organic processes, which may affect calculation results.
Research has shown that aquatic photosynthesis uses dissolved inorganic carbon to synthesize
organic carbon (Waterson and Canuel,2008;Tao et al.,2009), and this is also one of the factors
affecting the results. In addition, differences in basin surface water and groundwater proportions
controlled by geological landform could also affect the calculation results.
To sum up, the calculation results for carbon sink discharge from karstification using
watershed monitoring data in areas limited to a dominant single LUCC type may differ in a small
watershed where geomorphology, hydrology, and land use cover are different. This is one of the
reasons why there is such a large deviation in China's total carbon sink discharge estimated by
using carbon sink data from a single watershed in a karst region. Therefore, considering the
diversity of landform types and surface covers in the southwestern karst area, it is important to
develop a monitoring network in different topographical and surface cover regions, using a variety
of monitoring technologies to improve the accuracy of karst carbon sink estimates.
**5. Conclusion**
It is important basic significance to determine the main factors that affect the karst
geological carbon sink and understand the mechanism of their effects on the karst geological
carbon sink. Through the contrast analysis of flow, bicarbonate ion concentrations and carbon sink



discharge between the different sites in three stations located upstream, midstream and downstream of Houzhai basin, respectively, we analyzed the reasons for the difference of flow, bicarbonate ion concentrations and carbon sink discharge. The preliminary conclusions are as follows: (1) The carbon sink discharge was mainly controlled by the flow of each site, and LUCC type has important effects on the bicarbonate ions concentrations in each site. (2) The large difference in flow among sites did not lead to significant differences in bicarbonate ion concentrations in the sites, showing that the rapid increase in flow only has a partial dilution effect on ion concentrations. Due to the high speed and stability of chemical carbonate weathering, bicarbonate ion concentrations did not change significantly, and thus did not affect carbon sink discharge. (3) For different LUCC conditions, if runoff is stable, the influence of flow variation on ion concentration will be less than the effects of chemical carbonate weathering **by** different environmental conditions (comparison of LHT and LG results is 150%) on bicarbonate ion concentrations. However, if runoff increases significantly, the impact of runoff variation on bicarbonate ions will be greater than the effects of chemical carbonate weathering **by** different environmental conditions (comparison results of LHT and MSK).

In addition, this study without considering the proportional distribution problem of the surface and underground runoff in catchment area of each monitoring sites, which may have influence on the results. Therefore, it is necessary to monitor runoff and bicarbonate ions of the surface and underground simultaneously, but unfortunately the monitoring data we used did not achieve it.

### Acknowledgements

This work was partially supported by the National Natural Science Foundation of China (No.41371045) and the National Key Technology R&D Program of the Ministry of Science and Technology of China(No.2014BAB03B001).

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
