# Peer review of "Differences and influencing factors for underground water carbon uptake by karsts in Houzhai Basin, southwest China"

_Solid Earth, 2016_

## Referee Comment (RC1) · Anonymous Referee #1 · 17 Mar 2016

The paper in its present form is meaningless. I am sorry to tell that, but the following points are not clear in the paper, making it unreadable.

1) Carbonate dissolution can be considered a carbon uptake if (and only if) the dissolved carbonate does not precipitate again within several thousands of years after it was dissolved. This point is not discussed at all in the paper and it is simply assumed that the whole carbonate dissolution fulfills these conditions.

2) Generally, the flux of dissolved carbonate out of a given catchment area is approached through the integration of the instantaneous carbonate flux, i.e. the carbonate concentration multiplied by the water flux. Because discharge rate and chemistry varies much and quickly at karst springs, the measurements of carbonate concentration and discharge rate must be frequent enough to make a reasonable approximation of the real carbonate (carbon) flux. For catchment areas with a size between 50 and 100 km2 a daily value is seen as a minimal frequency. In the paper, the frequency is about 5 days inducing a real uncertainty in this result. A further problem is that the formula given for the calculation (equation 4) suggests that the authors used the seasonal average carbonate concentration multiplied by the instantaneous flow rate, inducing a supplementary bias.

3) The discharge rates given for the three measurement stations are very unclear: a) the measurement frequency is not given, neither the measurement expected accuracy; b) Table 1 gives numbers ranging between 2.67 and 21.83, with no indication of the unit, and just below graphics give values for the same stations ranging between 70 and 450 m3/s, but the sames numbers are given in m3 in the text. it is very confusing. Even more confusing is the fact that the average runoff given for the whole year seems to be the sum of the wet and dry season average runoff values, instead of being an intermediate (average) value.

4) According to the rainfall quantity given in the introduction and to the size of the catchment area, discharge rates given (even in table 1) are much too high. The total rainfall (1300 mm) over a surface area of 85 km2 gives a maximal annual average flow rate of 3.5 m3/s. Very confusing. What are the catchment areas of the three measurement stations? How were they determined?

5) For carbon flux, the transformation of the data measured in the field into the average data presented in the paper (§3.2) is quite uncertain. The measured data must be presented. The reader (or reviewer) could thus verify if transformations are correct.

6) The main conclusion is that most carbon is exported during the rainy season. The data presented in the paper (in its present form) enclose so much uncertainty that this result cannot by considered as supported by the data. By the way, if carbonate concentration decreases of about 10 to 20% during the wet season compared to the

dry one, and that discharge rate increases by 50%, it is quite clear that the exported carbonate will be higher during the wet season. The paper by Gremaud et al. is a nice example of such a situation.

7) English is poor, the structure of the paper is not clear and logical, figures 5 to 7 are just unnecessary replicates of figures 1 to 3, figure captions are not sufficient.

8) Discussion and conclusions cannot be understood.

I can only expect that the authors wrote their paper in Chinese language and that the translation was done by a person, who doesn't understand the subject. Eventhough, the original paper could not be a good one because there are real mistakes in the method applied and a real lack of critical sense on the presented data.

---

## Referee Comment (RC2) · Anonymous Referee #2 · 5 Apr 2016

In this manuscrip, authors analysis the influencing factors of chemical weathering and flow on karst carbon sink. The author accumulated a large amount of data, which is reliably for the discussion. This study has a certain guiding significance for carbon sink research. However, there are some erros listd as follow that should be corrected. After they are modifed, I think this mansucript can be published. (1) Runoff and chemical weathering were considered in this article, but what about the temperature, plants, aquatic organism, etc. At least, the factors should be explained in the introduction section. (2) Flow and ion concentration change and its effects on carbon sink, but hydrodynamic condition that control the cantact time of water-rock, may have more important influence on the carbon sink. So the results of the article should be deep

analyis. (3) The quality of the ms is poor in English express. I strongly suggest the authors to polish the English. There are many professional editing services.

Please also note the supplement to this comment:
http://www.solid-earth-discuss.net/se-2016-37/se-2016-37-RC2-supplement.pdf

---

## Author Comment (AC1) · 3 May 2016

The Summary of Revision for the SE Manuscript No.: se-2016-37 R1. Manuscript Title: Differences and influencing factors related to underground water carbon uptake by karsts in the Houzhai Basin, southwestern China

Thank you for reading and reviewing our manuscript. Your comments will definitely help us improve the manuscript to a higher scientific level. We revised our manuscript according to your comments and it contains all the changes to be visible. The points mentioned by the reviewers will be discussed below. We added some contents in blue and deleted some contents using revision mode in Word. Besides, some small revisions we made were not showed in Word.

[Figure]

Please see specific amendments in the revised manuscript.

Our reply note as follows:

Comment 1: Runoff and chemical weathering were considered in this article, but what about the temperature, plants, aquatic organism, etc. At least, the factors should be explained in the introduction section.

Reply1: The reviewer's opinion is very professional. Temperature, vegetation and aquatic organisms have important effects on the karst carbon sink, especially the carbon sequestration by aquatic organisms is the focus of the current research. Due to the limitation of data acquisition time and data acquisition means, our current calculation cannot well reflect the temperature, especially the effects of aquatic organisms on watershed carbon sequestration. Therefore, we added some related contents in the introduction according to the reviewer's comment.

Please see specific amendments in the revised manuscript. (Line 49-56, Page 2); (Line 66-79, Page 2-3 ); (Line 102-104, Page 3).

Comment 2: Flow and ion concentration change and its effects on carbon sink, but hydrodynamic condition that control the contact time of water-rock, may have more important influence on the carbon sink. So the results of the article should be deep analyze.

Reply 2: As the reviewer's comment, the water-rock contact time and temperature may have an important influence on carbon sink, however, we cannot obtain relevant data due to the limitation of our observation conditions. But we also attempt to add some contents in discussion part. Besides, the spatial resolution of the data we used is not compatible with the requirements, which is the direction we need to pay attention to in the future. At the same time, we follow the reviewer's comments and strengthen the discussion of the results in the manuscript.

Please see specific amendments in the revised manuscript. (Line 181-182 and Line

185-192, Page 6) ; (Line 197-199 and Line 205, Page 7); (Line 223, Page 8); (Line 240-242, Page 9). (Line 247-252, Page 10). (Line 263-266, Page 10). (Line 330-335, Page 12-13). (Line 337-339, Page 13). (Line 360-368, Page 13).

Comment 3: The quality of the ms is poor in English express. I strongly suggest the authors to polish the English. There are many professional editing services.

Reply 3: According to the reviewer's comments, we asked the professional editing institutions to polish the manuscript.

Attached please find the documentary evidence of article polishing and Response to reviewers' comments, Revised Manuscript.

Please also note the supplement to this comment:
http://www.solid-earth-discuss.net/se-2016-37/se-2016-37-AC1-supplement.zip

---

## Author Comment (AC2) · 3 May 2016

The Summary of Revision for the SE Manuscript No.: se-2016-37 R1. Manuscript Title: Differences and influencing factors for underground water carbon uptake by karsts in Houzhai Basin, southwest China

Thank you for reading and reviewing our manuscript. Your comments definitely help us improve the manuscript to a higher scientific level. The authors revised our manuscript according to your comments and it contains all the changes to be visible. The points mentioned by the reviewers will be discussed below. We added some contents in blue and deleted some contents using revision mode in Word. Besides, some small revisions we made were not showed in Word. Please see specific amendments in the

revised manuscript.

Our replies are as follows:

Comment 1:

Carbonate dissolution can be considered a carbon uptake if (and only if) the dissolved carbonate does not precipitate again within several thousands of years after it was dissolved. This point is not discussed at all in the paper and it is simply assumed that the whole carbonate dissolution fulfills these conditions.

Reply 1:

The authors want to apologize that the authors did not clearly introduce the latest research results of the stability of karst geological carbon sink, and now the authors added some related research in the introduction section. In addition, there are many research findings of karst geological carbon sink, and related research has clearly pointed out that it's not a simple cycle where the carbon dioxide was released again through the precipitation of calcium carbonate after tens of thousands of years. Because the numerous micro-organisms in karst ecosystem could use carbonate ions in their metabolic process, the carbon sink was formed permanently.

(Line 49-56, Page 2); (Line 66-79, Page 2-3); (Line 102-104, Page 3).

The relevant research findings can be referred in the following papers:

(1) Yan, J. ,Wang, Y. P. ,Zhou, G. ,Li, S. ,Yu, G. and Li, K.: Carbon uptake by karsts in the Houzhai Basin, southwest China, Journal of Geophysical Research, 116, doi: 10.1029/2011JG001686, 2011. (2) Yan, J. ,Wang, W. ,Zhou, C. ,Li, K. and Wang, S.: Responses of water yield and dissolved inorganic carbon export to forest recovery in the Houzhai karst basin, southwest China, Hydrological Processes, 28, 2082-2090, doi: 10.1002/hyp.9761, 2014. (3) Yan, J., Wang, Y.P., Zhou, G., Li, S., Yu, G., Wang, S., 2012. Reply to comment by François Bourgeset al. on Carbon uptake by karsts in the Houzhai Basin, southwest China". J GEOPHYS RES 117, doi:

10.1029/2012JG002060. (4) Jiang, Z. C. and Yuan, D. X.: CO2 source-sink in karst processes in karst areas of China, Episodes, 21, 33-35, 1999. (5) Liu, Z. and Zhao, J.: Contribution of carbonate rock weathering to the atmospheric CO2 sink, Environmental Geology, 33, 1053-1058, doi: 10.1007/s002549900072, 2000. (6) Lian, B. ,Yuan, D. X. and Liu, Z. H.: Effect of microbes on karstification in karst ecosystems, Chinese Science Bulletin, 56, 2158-2161, 2011. (7) Liu, Z. ,Dreybrodt, W. and Wang, H.: A new direction in effective accounting for the atmospheric CO2 budget: Considering the combined action of carbonate dissolution, the global water cycle and photosynthetic uptake of DIC by aquatic organisms, Earth-Science Reviews, 99, 162-172, doi: 10.1016/j.earscirev.2010.03.001, 2010.

Comment 2:

Generally, the flux of dissolved carbonate out of a given catchment area is approached through the integration of the instantaneous carbonate flux, i.e. the carbon at concentration multiplied by the water flux. Because discharge rate and chemistry varies much and quickly at karst springs, the measurements of carbonate concentration and discharge rate must be frequent enough to make a reasonable approximation of the real carbonate (carbon) flux. For catchment areas with a size between 50 and100 km2 a daily value is seen as a minimal frequency. In the paper, the frequency is about 5 days inducing a real uncertainty in this result. A further problem is that the formula given for the calculation (equation 4) suggests that the authors used the seasonal average carbonate concentration multiplied by the instantaneous flow rate, inducing a supplementary bias.

Reply 2:

The reviewer's comment is very accurate. Within the time of data collection, , the manual sampling method was adopted instead of the automatic monitoring equipment due to the money shortage in the research area. Manual sampling is time-consuming, high-cost, and the errors caused by manual sampling have more uncertainty than that

by the automatic monitoring equipment. Considering the limited conditions and actual situation of the ecological environment of the basin, the authors are only able to monitor the data twice a day. But both the continuous observation data the authors used and the discontinuous observation data without being used show that the bicarbonate ion concentration of underground springs was relatively stable in the basin.

The Houzhai Basin is located in the monsoon climate region that is the agriculture area as well. The seasonal changes of agricultural production significantly affect the flow, and the diurnal variation of the flow is obvious. Therefore, in order to further reduce the uncertainty of results, the authors adopt the average bicarbonate ion data of the dry season, the rainy season and the whole year. In terms of the reviewer's comment that the unit watershed area is generally 50-100km2,the literature showed that the catchment area in the same and similar studies was much smaller than that of the site in our paper, for example, the watershed area of the Chenqi basin is1.5km2.

Please see specific amendments in the revised manuscript.

(1) Zhao, M, Zeng, C, Liu, Z,Wang, S. 2010. Effect of different land use/land cover on karst hydrogeochemistry: A paired catchment study of Chenqi and Dengzhanhe, Puding, Guizhou, SW China. Journal of Hydrology 388: 121-130. DOI: 10.1016/j.jhydrol.2010.04.034 (2) Bourges F, Genthon P, Mangin A, et al. Microclimates of l'Avend'Orgnac and other French limestone caves (Chauvet, Esparros, Marsoulas)[J]. International Journal of Climatology, 2006, 26(12): 1651-1670. DOI:10.1002/joc.1327

Comment 3:

The discharge rates given for the three measurement stations are very unclear: a) the measurement frequency is not given, neither the measurement expected accuracy; b) Table 1 gives numbers ranging between 2.67 and 21.83, with no indication of the unit, and just below graphics give values for the same stations ranging between 70 and450 m3/s, but the sames numbers are given in m3 in the text. it is very confusing.

Even more confusing is the fact that the average runoff given for the whole year seems to be the sum of the wet and dry season average runoff values, instead of being an intermediate (average) value.

Reply 3:

a) As mentioned in the reply 2, the flow measurement was restricted by objective conditions as well. The flow data is converted from water level through manual monitor that is a basic method and most commonly used in hydrology monitoring field. Flow data were monitored twice a day:8:00 AM and 8:00 PM. The average of the two flow data was considered as flow data for one day. The description of flow data monitoring is added in the paper. The authors appreciate the reviewer's points. Please see specific amendments in the revised manuscript. (Line 157-160, Page 5).

b) The content of table 1 is the standard deviation of the flow, ion concentration and carbon flux in dry, wet season and a whole year, respectively, in order to analyze the stability of the above variables in the site. According to the flow data of Table 1 and Figure 2, 3,4 & 5 of every site, the author's original purpose is to calculate the average flow for every site in dry, wet season and a whole year. However, the flow data was not divided by the time duration because of the author's mistake, leading to the errors in the original manuscript. The author has revised the manuscript. the author sincerely apologize for the mistake and appreciate that the reviewer pointed out the mistake timely.

Please see specific amendments in the revised manuscript. (Line 212-214, Page 7), Table 1. (Line 214-294, Page 7-11), Figure 2, 3,4 & 5.

Comment 4:

According to the rainfall quantity given in the introduction and to the size of the catchment area, discharge rates given (even in table 1) are much too high. The total rainfall (1300 mm) over a surface area of 85 km2 gives a maximal annual average flow rate

of 3.5 m3/s. Very confusing. What are the catchment areas of the three measurement stations? How were they determined?

Reply 4:

a) The Houzhai basin is located in the typical karst area in Southwestern China and has strong karstification. The water system in the basin is mainly composed of the underground river and the surface river. Correspondingly, the discharge is composed of the surface and underground as well. The basin is located in the typical monsoon climate region, and the rainfall in rainy season can reach 80% of the whole year. However, because of the characteristics of underlying surface in the karst basin, the flood formed during the rainy season was mostly discharged by the surface river system. The underground water is relatively stable and small because the supply water is mainly from underground aquifer fissures and solution fissures.

b) The catchment areas of three sites were obtained by previous research results. The boundary and the catchment areas of the basin were determined when the basin was planned and constructed in 1980,and the monitoring station construction took full account of the representative of the sites. At the same time, a large number of research results have been published, and some of them are related to the catchment area of the three sites. Wang et al.,2010reached one of the most comprehensive research results, so the authors directly use the results of the study and mark it in the manuscript.

Please see specific amendments in the revised manuscript.

Comment 5:

For carbon flux, the transformation of the data measured in the field into the average data presented in the paper (§3.2) is quite uncertain. The measured data must be presented. The reader (or reviewer) could thus verify if transformations are correct.

Reply 5:

The authors use a lot of data and it is impossible to show one by one in the manuscript.
Besides, it is not required to show all the data but the most important result when publishing science papers. Meanwhile, as the authors can see from the formula, the calculation is simple and the authors double-checked the data. Additionally, there is a verbal agreement between the institution that provided the monitoring data and us, the authors cannot offer the original data to others. Therefore, I want to apologize for not providing the original data.

Comment 6:

The main conclusion is that most carbon is exported during the rainy season. The data presented in the paper (in its present form) enclose so much uncertainty that this result cannot be considered as supported by the data. By the way, if carbonate concentration decreases of about 10 to 20% during the wet season compared to the dry one, and that discharge rate increases by 50%, it is quite clear that the exported carbonate will be higher during the wet season. The paper by Gremaud et al. is a nice example of such a situation.

Reply 6:

The conclusion of the manuscript is based on our calculation. But spatial resolution of the monitoring data the authors used is low and the authors cannot obtain the similar conclusions with those of Gremaud et al. The purpose of our study is to explain the reasons that the research results by estimating the carbon flux in karst regions of China were different and to contribute to better promote development of related research in the future, through analyzing the reasons of the difference of carbon flux in every site. In addition, the instability and heterogeneity of karst system may lead to significant differences of different areas in the results, so the calculation from the single site data may be different.

Please see specific amendments in the revised manuscript.

Comment 7:

English is poor, the structure of the paper is not clear and logical, figures 5 to 7 arejust unnecessary replicates of figures 1 to 3, figure captions are not sufficient.

Reply 7:

According to the reviewers' comments, the authors polished the manuscript. At the same time, the purpose of the study is to make a comparative analysis of the same variables in each site, so as to determine the effects of different variables on the calculation. In addition, the figures 5 to 7 also help the readers to better understand the manuscript, rather than simply repeat the information from figures 1-3.

Please see specific amendments in the revised manuscript.

Comment 8:

Discussion and conclusions cannot be understood.

Reply 8:

In the discussion part, based on the calculation, the authors discuss the reason of the difference of carbon sink flux in every site from the relatively macroscopic view of the different environment of catchment area in the basin. As the reply of the reviewer's comment 6, according to our results, the authors cannot propose the similar conclusions as Gremaud et al research results. In addition, in order to further improve the quality of the manuscript and help the readers and reviewers easy to understand, the authors revised and polished the manuscript, specially the discussion and conclusion parts.

Please see specific amendments in the revised manuscript.

(Line 239-391, Page 9-15).

Finally, thanks again for the reviewer's hard working. Due to the limited academic ability of the author, the manuscript does not meet the reviewer's requirement. The author needs to learn more and improve himself in future. However, as a beginner in

academic field, the author hopes to get more encouragements and supports from the reviewer.

Please also note the supplement to this comment:
http://www.solid-earth-discuss.net/se-2016-37/se-2016-37-AC2-supplement.zip